# Prevalence, infection intensity, and risk factors of soil-transmitted helminthiasis and intestinal schistosomiasis among schoolchildren in Southern Ethiopia

Fisseha Bonja[1,2]*, Berhanu Erko[3], Musa Mohammed Ali[1], Bineyam Taye[4,5], Hagos Ashenafi[3]

1 College of Medicine and Health Sciences, School of Medical Laboratory Sciences, Hawassa University, Hawassa, Ethiopia, 2 Department of Microbiology, Immunology and Parasitology, College of Health Sciences, Addis Ababa University, Addis Ababa, Ethiopia, 3 Aklilu Lemma Institute of Health Research, Addis Ababa University, Addis Ababa, Ethiopia, 4 Department of Biology, Colgate University, Hamilton, New York, United States of America, 5 Global Public Environmental Health, Colgate University, Hamilton, New York, United States of America

* fisseha.bonja@aau.edu.et

## Abstract

### Background

Soil-transmitted helminthiasis (STH) and intestinal schistosomiasis are widespread and prevalent in tropical regions due to poor sanitation and limited healthcare access. Despite control efforts, localized epidemiological data remain essential for effective intervention. This study aimed to assess the prevalence of STH and associated factors among schoolchildren in Hawella Tulla sub-city, Hawassa City, Sidama National Regional State, Ethiopia.

### Method

A school-based cross-sectional study involving 394 primary schoolchildren was conducted from June to December 2023. Data were collected using an interviewer-administered questionnaire. Infection with the parasite was diagnosed by Kato–Katz microscopy. Bivariate and multivariable logistic regression were used to identify factors associated with the STH infections.

### Results

The combined prevalence of soil-transmitted helminths (STHs) and intestinal schistosomiasis was 42.6% (168/394; 95% CI: 37.8–47.6). STHs constituted the predominant infection, detected in 41.1% of children (162/394; 95% CI: 36.3–46.0), while intestinal schistosomiasis was found in 3.3% (13/394; 95% CI: 1.9–5.4). The most prevalent parasite was *Ascaris lumbricoides* (29.2%), followed by *Trichuris trichiura* (17.8%) and hookworms (5.1%). Schoolchildren who reported consuming unwashed

**Data availability statement:** The authors confirm that all data underlying the findings are fully available without restriction. All relevant data are within the paper and its Supporting information files.

**Funding:** The contribution of MMA was partially supported by Hawassa University, the NORAD Medium-Scale Research Support program. The contribution of HA and FBG was partially supported by Addis Ababa University for data collection. The funders had no role in the study design, data collection and analysis, decision to publish, or preparation of the manuscript.

**Competing interests:** The authors have declared that no competing interests exist.

fruits or vegetables (AOR: 2.89; 95% CI: 1.73–4.85), swimming or bathing in streams or lakes (AOR: 2.23; 95% CI: 1.31–3.79), and not receiving deworming treatment (AOR: 2.12; 95% CI: 1.26–3.56) were significantly more likely to be infected with soil-transmitted helminths (STHs).

## Conclusion

Overall, soil-transmitted helminthiasis (STHs) and *Schistosoma mansoni* remain significant public health concerns in the study area, affecting more than one-third of schoolchildren. Infections were strongly associated with poor hygiene practices, inadequate deworming coverage, and exposure to contaminated water sources. This highlights the need for improved hygiene, health education, and deworming programs.

## Author summary

Soil-transmitted helminthiasis and intestinal schistosomiasis remain major health problems in areas with poor sanitation, frequent contact with contaminated water, and limited access to clean water. In Ethiopia, these infections persist, particularly in rural communities where open defecation, frequent contact with contaminated water, and low hygiene awareness are common. This study investigated the prevalence and associated factors of STHs and *S. mansoni* infections among schoolchildren in Hawella Tula Rural District, Southern Ethiopia. Over 40% of children were infected with at least one parasite, with *A. lumbricoides* being the most prevalent, followed by *T. trichiura.* Infections were strongly linked to eating unwashed vegetables, swimming or bathing in contaminated water, and a lack of deworming treatment. Poor sanitation and limited access to safe water further increased the risk of infection. These findings indicate that soil-transmitted helminths (STH) and *Schistosoma mansoni* infections remain major public health challenges despite ongoing national control efforts. Strengthening school-based deworming programs, enhancing hygiene education, reducing exposure to unsafe water, and expanding access to safe water and sanitation are critical to safeguarding children's health in this region.

## Background

Soil- transmitted helminthiasis (STH), caused by *Ascaris lumbricoides*, *Trichuris trichiura*, and hookworms (*Ancylostoma duodenale and Necator americanus*), is a major public health concern in regions with poor sanitation and inadequate access to clean water [1–4]. In addition to STHs, intestinal schistosomiasis caused by *Schistosoma mansoni* is also prevalent in developing regions [5]. Globally, 4.5 billion people are at risk of STH infections, 1.5 billion infected people or 24% of the world's population [6], with an estimated number of people known to be infected globally is

807-1,121 million with ascariasis, 604–795 million with trichuriasis, 576–740 million with hookworm infection, and 240 million people infected with schistosomiasis [7]. In sub-Saharan Africa, about 198, 192, 173, and 162 million people are infected with hookworms, schistosomes, *Ascaris lumbricoides,* and *Trichuris trichiura*, respectively [8]. School children are particularly vulnerable to STH infections, primarily through contact with contaminated soil, poor hand hygiene, and geophagia [9,10]. Chronic STH infection contributes to malnutrition and long-term impairments in cognitive and physical development, adversely affecting educational outcomes and well-being [11,12]. As a core public health intervention, the WHO has recommended deworming treatment to all school-aged children for control of STH infections across high-risk populations in developing countries [13]. Mass deworming consists of a single dose of an anthelmintic medication (400 mg albendazole or 500 mg mebendazole), either annually in areas with STH infection prevalence of 20–50%, or biannually in areas with prevalence over 50% [14]. In Ethiopia, STHs are widely distributed, with an estimated 96.7 million people living in STH-endemic areas; about 27.7 million of these are school-aged children [15]. The Ethiopian Federal Ministry of Health launched a mass deworming program to treat more than 80% of at-risk school-aged children in all endemic regions [16]. This further supports the country's efforts to achieve SDGs targeting STH infections under Goal 3, aiming to reduce morbidity and improve health, especially in children, as part of NTD control [17–19]. Despite various efforts to reduce the burden of soil-transmitted helminth (STH) infections and schistosomiasis in Ethiopia, the prevalence of these infections has consistently remained above the WHO low-prevalence threshold. Various efforts are made to reduce morbidity among preschool-aged and school-aged children by lowering the prevalence of moderate- or heavy-intensity infections to less than 1% in school-aged children [13]. The persistent burden is driven by factors such as rapid reinfection after treatment [19], entrenched environmental exposure risks, and complex socio-demographic determinants [20,21].Together, these factors sustain transmission cycles and impede progress toward elimination targets. Therefore, continuous monitoring of socio-demographic, environmental, and lifestyle characteristics associated with increased STH risk is essential, and tailoring interventions to the specific needs of each community remains critical.

This study provides evidence on the prevalence and distribution of soil-transmitted helminthiasis (STHs), caused by *Ascaris lumbricoides*, *Trichuris trichiura*, and hookworms (*Ancylostoma duodenale* and *Necator americanus*), as well as intestinal schistosomiasis, caused by *Schistosoma mansoni.* We hypothesize that specific socio-demographic and environmental factors, such as household sanitation, water access, and parental education, are significantly associated with the risk of STH infection among school-aged children.

## Methods

### Ethics statement

Ethical clearance was obtained from Addis Ababa University Aklilu Lemma Institute of Health Research-Institutional Research Ethics Review Committee. Permission was obtained from sub city health and education department to each selected school. All participants were well informed about the purpose and the procedures of the study. All responses were kept confidential and anonymous. Participation was fully voluntary; assent was obtained from each child and formal written consent was also obtained from the parent/guardian.

### Study setting and design

The study was conducted among school-aged children in Hawella Tulla sub-city, Hawassa city, Sidama National Regional State, Southern Ethiopia (Fig 1). Hawassa city is administratively divided into 8 sub-cities. The city is located at approximately 7°03′N and 38°28′E, along the northern shore of Lake Hawassa in the Great Rift Valley, and serves as the capital of the Sidama Region. According to recent projections, Hawassa has an estimated population of more than 550,000, with a large proportion of children, adolescents, and working-age adults. The demographic structure shows that nearly two-thirds of the population is under 25 years of age, suggesting a substantial number of school-age children. The Sub city district has an agrarian economy focused on coffee, maize, livestock, and some Khat production. The Hawela Tula

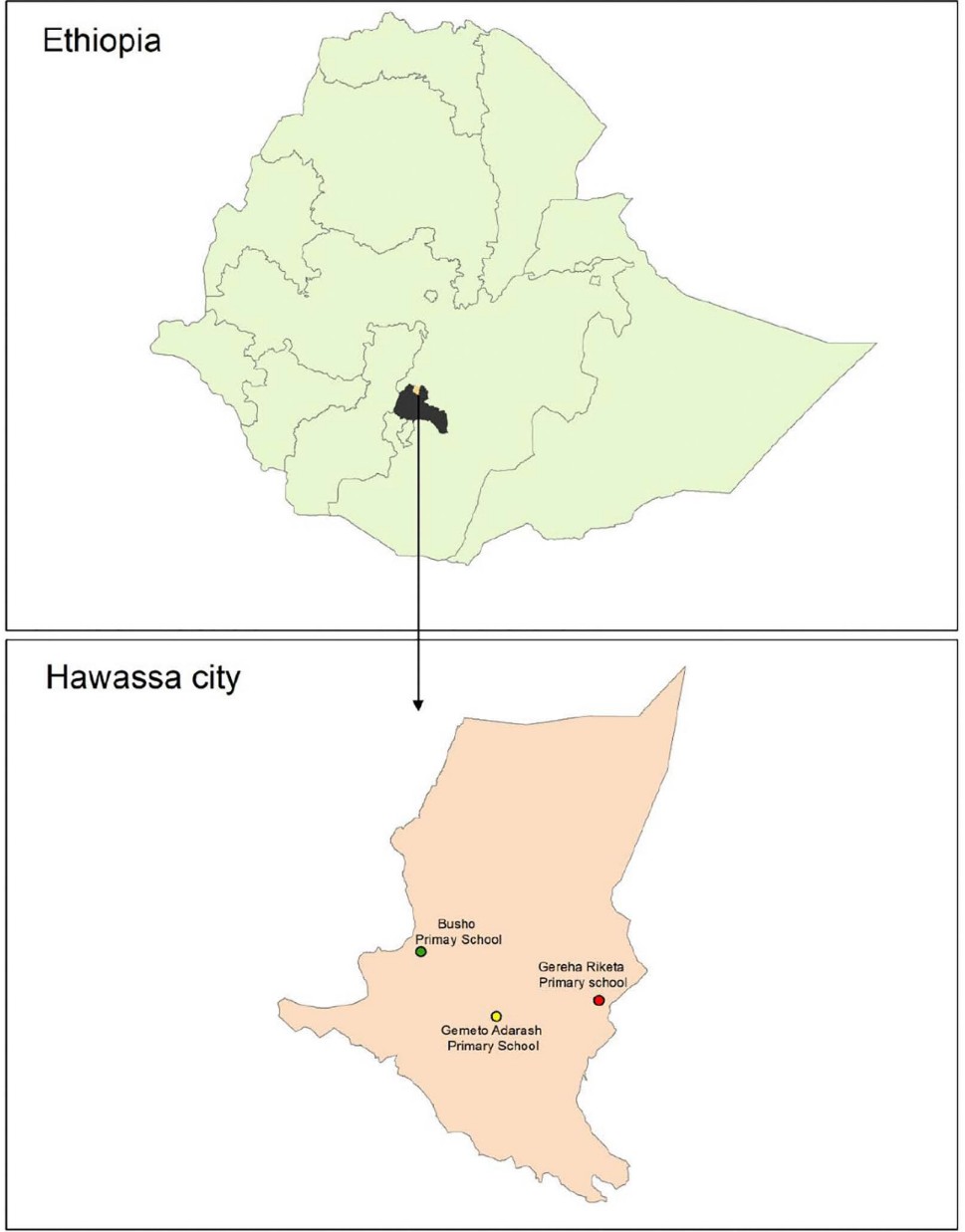

**Fig 1. Map of study area.** The country or region border shapefile is obtained from website: https://data.humdata.org/dataset/cod-ab-eth.

sub-city is home to three governmental primary schools and 2 private schools, with 11,134 students enrolled across 3 governmental schools in the 2023 academic year. The main staple food is Enset (false banana), and coffee and maize are key crops grown in the area [22]. The study was carried out from June to December 2023.

## Study population and eligibility criteria

This school-based cross-sectional study focused on only primary schoolchildren (5–18 years) in Hawella Tula Rural district, Hawassa, Sidama Regional State, Ethiopia. The study site was selected based on the burden of soil-transmitted

helminth infections and schistosomiasis, and all three primary schools within the area were included in the study. The study included those who had lived in the area for at least six months and excluded children with recent anti-helminthic treatment or a history of intestinal parasitic treatment within two weeks, as well as students unable to provide assent due to serious illness.

## Study variables

**Outcome variable:** Soil-transmitted helminth (STH) and S. mansoni infections.

**Exposure variables:** Socio-demographic and economic factors (child's age, sex, residence, parental education, occupation, family size, and household wealth) were considered. Environmental factors (soil contamination, poor sanitation, open defecation, proximity to water bodies) were taken into account. Behavioral aspects (barefoot walking, swimming in contaminated water, poor hand hygiene, improper food handling, lack of health awareness) were recorded. Additionally, WASH-related characteristics (access to clean water, availability of latrines, hand washing facilities, proper waste disposal) were assessed.

## Sample size and sampling procedures

The sample size (n) was calculated using the following single population proportion formula [23],
based on the assumption of previous STH prevalence (p) of 54% in schoolchildren reported from South Ethiopia [2], 95% Confidence interval (CI) (1.96), 5% margin of error (d), and adding 10% contingency.

Therefore, the required sample size was calculated and determined to be 420.

For the selection of the study participants, the sampling frames available in the district and, primary schools were used. The sample was allocated to all available schools. Then the study participants proportionally allocated to each school based on the number of children from 7–18. Finally, 420 study participants were selected by using Microsoft-Excel - generated random number (lottery method).

## Data collection tools and procedures

### Data collection tools

Data were collected through interviews using the Amharic-translated questionnaires. Socio-demographic factors and other determinants were assessed, with a pretest on 5% of the study population conducted two weeks prior to the actual study. The data were collected by trained data collectors who completed two days of training followed by competency assessments.

### *Schistosoma mansoni*

Stool samples were collected in clean, labeled plastic containers and examined using the Kato-Katz thick smear technique [24]. During the survey, stool samples were collected on-site and delivered to the laboratory within 10 minutes of defecation. A portion of each sample was examined immediately (within 10–30 minutes) to detect hookworms and other STH parasites, while the remaining portion was preserved for further analysis. A portion of the fecal sample was taken with a wooden spatula, sieved through a nylon screen, and transferred to a template on a microscope slide. The template hole was filled with the sieved sample, leveled, and covered with malachite green-glycerin-soaked cellophane squares. The slides were then examined under a microscope for STH eggs. A Kato-Katz thick smear was prepared from each sample and analyzed under a light microscope for hookworm eggs between 30 and 60 minutes after slide preparation [25]. A child was classified as positive for soil-transmitted helminths (STH) if eggs of *Ascaris  lumbricoides*, *Trichuris trichura*, or hookworms (*Ancylostoma duodenale or* or *Necator americanus NECATOR AMERICANUS*) were detected in the stool

sample. Intestinal schistosomiasis was confirmed when Schis*toma mansoni* eggs were identified. If no parasite eggs were observed in the processed slides, the child was classified as negative. Egg counts were recorded as eggs per gram (EPG) by averaging counts from two slides and multiplying by a conversion factor of 24. Infection intensity was classified according to WHO criteria. For *Ascaris lumbricoides*, light infection was defined as 1–4,999 eggs per gram (EPG), moderate as 5,000–49,999 EPG, and heavy as ≥50,000 EPG. For *TRICHURIS TRICHIURA*, light, moderate, and heavy infections corresponded to 1–999 EPG, 1,000–9,999 EPG, and ≥10,000 EPG, respectively. Hookworm infection intensity was categorized as light (1–1,999 EPG), moderate (2,000–3,999 EPG), and heavy (≥4,000 EPG). For *Schistosoma mansoni*, infection was considered light at 1–99 EPG, moderate at 100–399 EPG, and heavy at ≥400 EPG. For quality control, duplicate slides were prepared for each sample and examined by two different technicians. Negative samples were rechecked the same day to ensure accuracy [26,27].

## Data analysis

Data was analyzed using SPSS version 23. Descriptive analysis, including frequency distribution and percentage, was made to determine the prevalence of STH, to describe socio-economic, demographic, and other determinants. Bivariate logistic regression analysis was conducted to estimate crude odds ratios (CORs), and all factors with a p-value <0.25 were considered candidates for multivariable logistic regression to control for confounding effects. The Hosmer-Lemeshow goodness-of-fit test indicated adequate model fit ($\chi^2(8) = 6.34$, p = 0.597), which was used to assess whether the necessary assumptions for multiple logistic regression were met. Adjusted odds ratios (AOR) with 95% confidence intervals (CI) were used to measure the strength of the association between outcome variables (STH) and its associated factors. Finally, a p-value <0.05 was considered significant.

## Results

### Socio-demographic characteristics

A total of 394 participants were interviewed yielding an overall response rate of 94%. The mean age was 11.7 years (SD ± 2.3); 252 (64.0%) were aged ≤12 years, and 212 (53.8%) were females. Most participants 383, (97.2%) lived in rural areas. Over half of the mothers, 205 (52.0%), had primary education, 50 (12.7%) had no formal education, and 30 (7.6%) had completed college or higher education. Among fathers, 23 (5.8%) had no formal education, and 77 (19.5%) had completed college or above. Regarding children's schooling, 171 (43.4%) were in grades 5–6, and 112 (28.4%) in grades 1–4. Most mothers (315, 79.9%) were housewives, and 120 (30.5%) of the fathers worked in private business. Family size showed that 268 (68.0%) of the children came from families with ≤5 members, and 59(15.0%) had fathers who practice polygamy (Table 1).

### WASH and other related characteristics

Among 394 participants, regarding WASH and related characteristics, 315 (79.9%) households had piped water, 18.0% used public taps, and 2.0% relied on well water or hand pumps. At the school level, 51.8% of children had no access to water, while 48.2% had access to piped water. Regarding water treatment, 64.0% of households did not treat water before drinking, and 95.9% did not store drinking water separately. Water usage: 66.2% used one 20L jerrycan, and 30.5% used two. Half of the participants washed their hands after contact with soil, and 32.5% washed fruits and vegetables before eating. Additionally, 77.4% of the children swam or bathed in streams, or lakes (S1 Table).

Regarding health-related characteristics, 276 (70.1%) of schoolchildren had experienced illnesses in the past two weeks. The majority, 216 (54.8%), had constipation or diarrhea, while 177 (44.9%) had a cough. A significant number, 252 (64.0%), experienced abdominal pain, and 245 (62.2%) had loss of appetite. 375 (95.2%) were delivered vaginally, and 174 (44.2%) received deworming treatment (Table 2).

**Table 1. Socio-demographic and socio-economic characteristics of study participants.**

| Variables | Category | Frequency (%) |
|---|---|---|
| Age group in year | ≤ 12 | 252 (64) |
| | > 12 | 142 (36) |
| Gender | Male | 182 (46.2) |
| | Female | 212 (53.8) |
| Place of residence | Urban | 11 (2.8) |
| | Rural | 383 (97.2) |
| Mother's Education | No formal education | 50 (12.7) |
| | Primary | 205 (52) |
| | Secondary | 109 (27.7) |
| | College and above | 30 (7.6) |
| Father's Education | No formal education | 23 (5.8) |
| | Primary | 128 (32.5) |
| | Secondary | 166 (42.1) |
| | College and above | 77 (19.5) |
| Highest Education Level of Child (Grade) | 1-4 | 112 (28.4) |
| | 5-6 | 171 (43.4) |
| | 6-8 | 111 (28.2) |
| Mother's Occupation | Housewife | 315 (79.9) |
| | Gov't employee | 37 (9.4) |
| | Private business | 32 (8.1) |
| | Student | 2 (0.5) |
| | Others | 8 (2) |
| Father's Occupation | Farmer | 131 (330.2) |
| | Student | 7 (1.8) |
| | Private business | 120 (30.5) |
| | Daily Laborer | 37 (9.4) |
| | Gov't employee | 97 (24.6) |
| | Others | 2 (0.5) |
| Husband had polygamy | No | 335 (85) |
| | Yes | 59 (15) |
| Child's Living Condition | With parents | 387 (98.2) |
| | Relative | 7 (1.8) |

## Distribution of soil-transmitted helminthiasis and intestinal schistosomiasis

The combined prevalence of soil-transmitted helminthiasis (STH) and intestinal schistosomiasis was 42.6% (168/394; 95% CI: 37.8–47.6). STH infections were identified in 41.1% of participants (162/394; 95% CI: 36.3–46.0), while schistosomiasis was detected in 3.3% (13/394; 95% CI: 1.9–5.4). Among the STH species, *Ascaris lumbricoides* was the predominant parasite, affecting 29.2% of children (115/394; 95% CI: 24.9–33.8). This was followed by *Trichuris trichiura* at 17.8% (70/394; 95% CI: 14.2–21.8) and hookworm infections at 5.1% (20/394; 95% CI: 3.2–7.6). Infection with *Schistosoma mansoni* was observed in 3.3% of the study population (13/394; 95% CI: 1.9–5.4) (Fig 2). We also found moderate and light *A. lumbricoides* infections in 54 (13.7%) and 61 (15.5%) children, respectively, and heavy, moderate, and light *T. trichiura* infections in 2 (0.5%), 11 (2.8%), and 53 (13.5%) children, respectively (Fig 3).

**Table 2. Multivariate logistic regression associated with soil-transmitted helminths among study participants.**

| Variables | Category | STH infection | | COR (95% CI) | AOR (95% CI) | P-Value |
|---|---|---|---|---|---|---|
| | | Yes n (%) | No n (%) | | | |
| Maternal education | No formal education | 19 (38.0) | 31 (62.0) | 1 | 1 | |
| | Primary education | 91 (44.4) | 114 (55.6) | 1.302 (0.69, 2.46) | 1.321 (0.650, 2.685) | 0.442 |
| | Secondary & above | 52 (37.4) | 87 (62.6) | 0.975 (0.50, 1.90) | 1.059 (0.499, 2.251) | 0.881 |
| Washing fruits/vegetables before eating | No | 132 (49.6) | 134 (50.4) | 3.218 (2.002, 5.171) | 2.781 (1.650, 4.688) | <0.001* |
| | Yes | 30 (23.4) | 98 (76.6) | 1 | 1 | |
| Avoid swimming or bathing in rivers/ponds/streams/lake | No | 42 (47.2) | 47 (52.8) | 1.378 (0.856, 2.216) | 1.631 (0.960, 2.771) | 0.070 |
| | Yes | 120 (39.3) | 185 (60.7) | 1 | 1 | |
| Deworming treatment given by parents/health professionals | No | 137 (48.4) | 146 (51.6) | 3.228 (1.953, 5.336) | 2.912 (1.698, 4.996) | <0.001* |
| | Yes | 25 (22.5) | 86 (77.5) | 1 | 1 | |
| Water treatment before drinking | No | 117 (46.4) | 135 (53.6) | 1.868 (1.213, 2.876) | 1.619 (1.008, 2.598) | 0.046* |
| | Yes | 45 (31.7) | 97 (68.3) | 1 | 1 | |
| Household food insecurity | Food secure | 30 (33.3) | 60 (66.7) | 1 | 1 | |
| | Mild/Moderately | 108 (42.5) | 146 (57.5) | 1.479 (0.894, 2.449) | 1.586 (0.905, 2.782) | 0.107 |
| | Severely insecure | 24 (48.0) | 26 (52.0) | 1.846 (0.910, 3.744) | 2.735 (1.197, 6.252) | 0.017* |
| Height for Age | Normal height | 96 (35.4) | 175 (64.6) | 1 | 1 | |
| | Stunted | 66 (53.7) | 57 (46.3) | 2.111 (1.369, 3.255) | 2.127 (1.256, 3.600) | 0.0058* |
| BMI for Age | Normal weight | 101 (38.5) | 161 (61.5) | 1 | 1 | |
| | Under weight | 61 (46.2) | 71 (53.8) | 1.370 (0.897, 2.091) | 1.358 (0.799, 2.308) | 0.258 |

NB: * statistically significant on multivariate analysis p-value (<0.05), COR: crude odds ratio, AOR: adjusted odds ratio, CI: confidence interval, 1: reference.

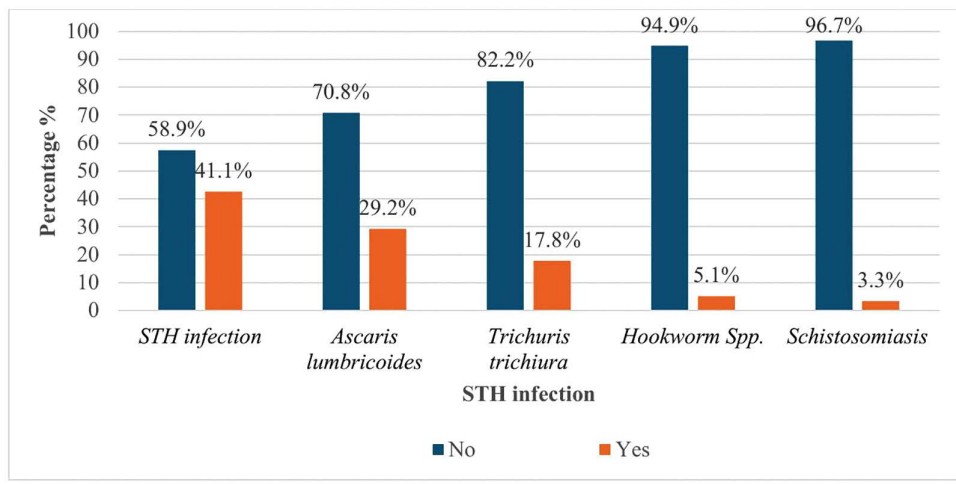

Note: - No: absence of parasite egg in stool sample. Yes: presence of parasite egg in stool sample.

**Fig 2. Prevalence of soil-transmitted helminths and *Intestinal schistosomiasis* among study participants.**

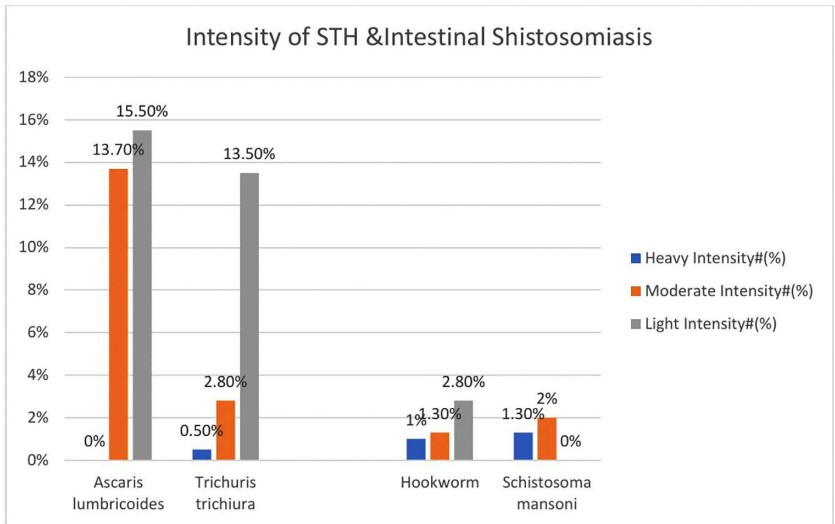

Note. Heavy intensity (for *Ascaris*>50,000 egg per gram, *Trichuris* >10,000 eggs per gram, *hookworm* (>4,000 eggs per gram, *Schistosoma xxx egg/gram),* Moderate intensity(for *Ascaris*5,000 – 49,999 egg per gram, *Trichuris* 1,000–9,999 eggs per gram, *hookworm* 2,000–3,999 eggs per gram, *Schistosoma xxx egg/gram),*Light intensity(for *Ascaris*1–4,999 egg per gram, *Trichuris* 1–999 eggs per gram,*hookworm*1–1,999 eggs per gram, *Schistosoma xxx egg/gram)*

**Fig 3. Intensity of soil-transmitted helminths and Intestinal schistosomiasis among study participants.**

### Factors associated with soil-transmitted helminthiasis among schoolchildren

In bivariate logistic regression, socio-demographic, economic, environmental, behavioral, and WASH-related factors with p < 0.25 were selected for multivariate analysis. The multivariate model identified significant predictors of *A. lumbricoides* and *T. trichiura* infections in schoolchildren (p < 0.05).In multivariate logistic regression, STH risk was higher among children who ate unwashed fruits/vegetables (AOR 2.89; 95% CI: 1.73–4.85), swam in streams/lakes (AOR 2.23; 95% CI: 1.31–3.79), had not been dewormed (AOR 2.12; 95% CI: 1.26–3.56), were from severely food-insecure families (AOR 2.87; 95% CI: 1.27–6.53), or were stunted (AOR; 2.08, 95% CI: 1.23–3.51) as compared with their counterparts (Table 2).

A separate logistic regression using *Ascaris lumbricoides* infection as the outcome identified similar risk factors to those for any STH infection, including lack of deworming treatment (AOR 2.34; 95% CI: 1.30–4.23), severe household food insecurity (AOR 2.90; 95% CI: 1.24–6.77), and stunting (AOR 1.72; 95% CI: 1.01–2.93) (Table 3).

### Discussion

The present cross-sectional study found a combined prevalence of soil-transmitted helminthiasis (STH) and intestinal schistosomiasis of 42.6%. STHs were the dominant infections, affecting 41.1% of children, whereas intestinal schistosomiasis accounted for 3.3%. Among the STH species identified, *ASCARIS LUMBRICOIDES* was most common (29.2%), followed by *TRICHURIS TRICHIURA* (17.8%) and hookworms (5.1%).

**Table 3. Multivariate logistic regression associated with *A. lumbricoides* among study participants.**

| Variables | Category | *Ascaris lumbricoides* | | COR (95% CI) | AOR (95% CI) | P-Value |
|---|---|---|---|---|---|---|
| | | Positive n (%) | Negative n (%) | | | |
| Residence | Urban | 6(54.5) | 5(45.5) | 1 | 1 | |
| | Rural | 109(28.5) | 274(71.5) | 3.02 (0.90, 10.09) | 2.35 (0.66, 8.43) | 0.190 |
| Age of Child | ≤ 12 | 64(25.4) | 188(74.6) | 1 | 1 | |
| | > 12 | 51(35.9) | 91(64.1) | 1.65 (1.06, 2.57) | 1.36 (0.82, 2.25) | 0.237 |
| Washing Fruits/Vegetables Before Eating | No | 94(35.3) | 172(64.7) | 2.79 (1.64, 4.74) | 2.24 (1.26, 3.98) | 0.006* |
| | Yes | 21(16.4) | 107(83.6) | 1 | 1 | |
| Had deworming treatment | No | 96(33.9) | 187(66.1) | 2.49 (1.43, 4.32) | 2.34 (1.30, 4.23) | 0.005 |
| | Yes | 19(17.1) | 92(82.9) | 1 | 1 | |
| Water Treatment Before Drinking | No | 84(33.3) | 168(66.7) | 1.79 (1.11, 2.88) | 1.58 (0.95, 2.63) | 0.081 |
| | Yes | 31(21.8) | 111(78.2) | 1 | 1 | |
| Household food insecurity | Food secure | 19(21.1) | 71(78.9) | 1 | 1 | |
| | Mild/Moderately insecure | 76(29.9) | 178(70.1) | 1.60 (0.90, 2.83) | 1.54 (0.83, 2.85) | 0.171 |
| | Severely insecure | 20(40) | 30(60) | 2.49 (1.17, 5.32) | 2.90 (1.24, 6.77) | 0.014* |
| Height for Age | Normal height | 67(24.7) | 204(75.3) | 1 | 1 | |
| | Stunted | 48(39) | 75(61) | 1.95 (1.24, 3.07) | 1.72 (1.01, 2.93) | 0.047* |
| BMI for Age | Normal weight | 70(26.7) | 192(73.3) | 1 | 1 | |
| | Under weight | 45(34.1) | 87(65.9) | 1.42 (0.90, 2.23) | 1.26 (0.72, 2.21) | 0.418 |

**NB:** * *statistically significant on multivariate analysis p-value (<0.05),* **COR:** *crude odds ratio,* **AOR:** *adjusted odds ratio,* **CI:** *confidence interval,* **1:** *reference.* Furthermore, a child who had no deworming treatment had an AOR of 4.64 (95% CI: 1.92, 11.21) and was more likely to be exposed to *T. trichiura* among schoolchildren than their counterparts (Table 4).

**Table 4. Multivariate logistic regression associated with *T. trichiura* among study participants.**

| Variables | Category | *Trichuris trichiura* | | COR (95% CI) | AOR (95% CI) | P-Value |
|---|---|---|---|---|---|---|
| | | Positive n (%) | Negative n (%) | | | |
| Washing Fruits/Vegetables Before Eating | No | 56(21.1) | 210(78.9) | 2.17 (1.16, 4.07) | 1.73 (0.89, 3.37) | 0.109 |
| | Yes | 14(10.9) | 114(89.1) | 1 | 1 | |
| Avoid swimming or bathing in streams/lake | No | 21(23.6) | 68(76.4) | 1.61 (0.91, 2.87) | 1.70 (0.93, 3.12) | 0.086 |
| | Yes | 49(16.1) | 256(83.9) | 1 | 1 | |
| Had deworming treatment | No | 64(22.2) | 219(77.4) | 5.11 (2.15, 12.19) | 4.64 (1.92, 11.21) | 0.001* |
| | Yes | 6(5.4) | 105(94.6) | 1 | 1 | |
| Water Treatment Before Drinking | No | 49(19.4) | 203(80.6) | 1.39 (0.80, 2.43) | 1.21 (0.67, 2.18) | 0.536 |
| | Yes | 21(14.8) | 121(85.2) | 1 | 1 | |
| Dietary diversity score | Good | 14(12) | 103(88) | 1 | 1 | |
| | Poor | 56(20.2) | 221(79.8) | 1.86 (0.99, 3.50) | 1.90 (0.98, 3.68) | 0.056 |
| Height for Age | Normal weight | 43(15.9) | 228(84.1) | 1 | 1 | |
| | Under weight | 27(22) | 96(78) | 1.49 (0.87, 2.55) | 1.23 (0.70, 2.17) | 0.471 |

**NB:** * *statistically significant on multivariate analysis p-value (<0.05),* **COR:** *crude odds ratio,* **AOR:** *adjusted odds ratio,* **CI:** *confidence interval,* **1:** *reference.*

The STH prevalence observed in this study is comparable to reports of 38.2% in southwest Nigeria [27], 46.7% in Jimma, Oromia, Ethiopia [28], and 39.0% Northwest Ethiopia [29]. However, it is lower than findings in Yirgacheffee, Southern Ethiopia (54%) [2], Mettu town, Southwest Ethiopia (84.4%) [3], Kola Diba Primary School in Northwest Ethiopia

(50.0%) [30], and Gara Riketa Primary School in Hawassa Tula Sub-City (67.7%) [4]. Conversely, it exceeds the 27.1% reported in Birbir town, Southern Ethiopia [31], and 36.1% in Rwanda [32]. These variations may reflect differences in environmental and geographic conditions, diagnostic techniques, sanitation and hygiene practices, deworming coverage, and the effectiveness of local health interventions. Additionally, climatic factors, urbanization, and behavioral practices such as walking barefoot and sanitation habits may also contribute to the observed differences [33]. Children who consumed unwashed fruits or vegetables were at greater risk of STH infection, consistent with studies conducted in southern Ethiopia and rural western Uganda [31,34]. The increased risk is biologically plausible, as unwashed produce may be contaminated with STH eggs originating from soil, human feces, or contaminated water sources [35].

This study also revealed that schoolchildren who swam or bathed in streams or lakes faced a higher risk of STH infection. Similar findings were also reported from southern Ethiopia [31] and north-west Ethiopia [36]. Some of the possible reasons could be the fact that schoolchildren can easily be exposed to contaminated water sources, which may contain helminth eggs or larvae from human and animal feces [35].

Furthermore, children who had not received deworming treatment were more likely to be infected, in agreement with findings from several regions of Ethiopia [2–4,36]. Lack of regular deworming may sustain transmission cycles and increase the risk of reinfection from contaminated environments [37,38].

The substantial burden of STH underscores the need to strengthen hygiene practices, improve access to safe water and sanitation, and maintain effective deworming programs. Inadequate sanitation, unsafe water, and limited preventive chemotherapy contribute to ongoing transmission and adverse health outcomes among schoolchildren [38].

## Limitations

Our findings should be interpreted considering several limitations. We used a cross-sectional study design to collect baseline data on children's parasite infection status and predictors. This design limits our ability to draw causal inferences about the relationships between parasite status and identified risk factors. Additionally, the low diagnostic sensitivity of the Kato-Katz (KK) technique is a widely recognized limitation, particularly when dealing with low-sensitivity infections, which could lead to underestimation of infection prevalence in epidemiological surveys.

## Conclusions

This study demonstrates that more than 40% of schoolchildren in Hawella Tula Rural District are infected with soil-transmitted helminths and *Schistosoma mansoni*, indicating a substantial and persistent burden in this endemic area. Key predictors of infection, including consumption of unwashed fruits and vegetables, bathing in contaminated water sources, and lack of deworming, highlight the important roles of environmental contamination, poor hygiene practices, and gaps in preventive interventions in sustaining transmission.

Integrated control strategies, including regular school-based deworming, targeted health education to promote hygiene and safe food handling practices, and provision of safe water, improved sanitation, hygiene (WASH) infrastructure, and strengthened annual mass drug administration (MDA) is recommended. By linking infection patterns to actionable interventions, this study provides policymakers and local health authorities with valuable evidence to prioritize resources and implement context-specific programs aimed at reducing transmission, improving child health outcomes, and advancing national and global goals for the control of neglected tropical diseases.

## Supporting information

**S1 Table. Raw SPSS data-Soil transmitted helminths among school age children.** (SPSS).
(SAV)

**S2 Table. English version questionnaire on soil transmitted helminths among school age children final.**
(DOCX)

**S3 Table. Amharic version questionnaire on soil transmitted helminths among school age children final.**
(DOCX)

## Acknowledgments

The authors gratefully acknowledge Addis Ababa University and Hawassa University for their provision of laboratory reagents. We also extend our sincere appreciation to the data collectors, school administrators, and study participants for their invaluable cooperation and contributions to this research.

## Author contributions

**Conceptualization:** Fisseha Bonja.

**Data curation:** Fisseha Bonja, Berhanu Erko, Musa Mohammed Ali, Bineyam Taye, Hagos Ashenafi.

**Formal analysis:** Fisseha Bonja.

**Funding acquisition:** Fisseha Bonja, Berhanu Erko, Musa Mohammed Ali, Bineyam Taye, Hagos Ashenafi.

**Investigation:** Fisseha Bonja.

**Methodology:** Fisseha Bonja, Berhanu Erko, Musa Mohammed Ali, Bineyam Taye, Hagos Ashenafi.

**Project administration:** Fisseha Bonja.

**Resources:** Fisseha Bonja, Bineyam Taye.

**Software:** Fisseha Bonja.

**Supervision:** Berhanu Erko, Musa Mohammed Ali, Bineyam Taye, Hagos Ashenafi.

**Validation:** Fisseha Bonja, Berhanu Erko, Musa Mohammed Ali, Bineyam Taye, Hagos Ashenafi.

**Visualization:** Fisseha Bonja.

**Writing – original draft:** Fisseha Bonja.

**Writing – review & editing:** Fisseha Bonja, Musa Mohammed Ali, Bineyam Taye, Hagos Ashenafi.

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
