## [Decision Letter · Decision Letter 0]

10 Nov 2025

Prevalence and intensity of soil-transmitted helminthiasis and associated factors among school children in Hawella Tula Rural district, Hawassa, Sidama Regional State, Ethiopia

Dear Dr. Geleto,

Thank you for submitting your manuscript to PLOS Neglected Tropical Diseases. After careful consideration, we feel that it has merit but does not fully meet PLOS Neglected Tropical Diseases's publication criteria as it currently stands. Therefore, we invite you to submit a revised version of the manuscript that addresses the points raised during the review process.

Please submit your revised manuscript within by Jan 09 2026 11:59PM. If you will need more time than this to complete your revisions, please reply to this message or contact the journal office at plosntds@plos.org. Please include the following items when submitting your revised manuscript:

We look forward to receiving your revised manuscript.

Kind regards,

Maria Y Pakharukova, Ph.D., D.Sc.

Academic Editor

Eva Clark

Section Editor

Shaden Kamhawi

co-Editor-in-Chief

Paul Brindley

co-Editor-in-Chief

**Journal Requirements:**

At this stage, the following Authors/Authors require contributions: Berhanu Erko, Musa Mohammed Ali, Bineyam Taye, and Hagos Ashenafi. Please ensure that the full contributions of each author are acknowledged in the "Add/Edit/Remove Authors" section of our submission form.

4) We notice that your supplementary Tables are included in the manuscript file. Please remove them and upload them with the file type 'Supporting Information'. Please ensure that each Supporting Information file has a legend listed in the manuscript after the references list.

5) In the online submission form, you indicated that "Raw data used for preparation of this manuscript can be obtained from corresponding author with resealable request.". All PLOS journals now require all data underlying the findings described in their manuscript to be freely available to other researchers, either

1. In a public repository

2. Within the manuscript itself

3. Uploaded as supplementary information.

**Reviewers' Comments:**

Reviewer's Responses to Questions

**Key Review Criteria Required for Acceptance?**

**Methods:**

-Are the objectives of the study clearly articulated with a clear testable hypothesis stated?

-Is the study design appropriate to address the stated objectives?

-Is the population clearly described and appropriate for the hypothesis being tested?

-Is the sample size sufficient to ensure adequate power to address the hypothesis being tested?

-Were correct statistical analysis used to support conclusions?

-Are there concerns about ethical or regulatory requirements being met?

Reviewer #1: The study objectives are generally clear, however, the cross-sectional design appears appropriate for assessing prevalence and associated factors. The study population, school-aged children in the selected district is relevant and appropriate for the objectives. Information on sample size determination is clearly provided. The statistical analyses used appear suitable for descriptive purposes. Finally, ethical considerations was described, including ethical approval, and adherence to institutional or national research ethics guidelines.

Reviewer #2: The method is well applied, although the sum of the parasitic infections is a matter that needs to be reconsidered.

Reviewer #3: - The study aims to assess the prevalence and intensity of STH and identify associated factors among schoolchildren, a topic relevant and important public health objective. However, there was no explicitly stated testable hypothesis.

- The study adopted a cross-sectional design, which is appropriate for estimating prevalence and identifying associations; however, this design limits causal inference.

- The population is well described as schoolchildren (5–18 years) and their caregivers residing in Hawella Tula Rural district for at least six months. However, it is unclear if the choice of site was linked to disease burden, epidemiology, or other risk factors; this rationale should be clarified. Also, the abstract does not mention the study site. Line 106 notes that "data were collected using pre-tested, structured, and translated questionnaires"—please specify the language(s) used for translation.

- Sample size calculation is based on a prior prevalence estimate (54%) and includes a contingency margin. Although 420 participants were targeted, only 394 were analyzed. The impact of this shortfall on statistical power should be discussed. Questions to address include:

i)How many schools were sampled?

ii)When and where were the samples analyzed? On-site or in a laboratory?

iii)Within how many hours of sample collection were analyses performed?

iv)What criteria were used for school (site) selection?

v)Were GPS coordinates collected for the schools? If yes, please provide a map showing their spatial distribution; if not, please clarify why.

vi)How were participants classified as STH or S. mansoni positive, and how was worm intensity measured?

vii)What is the mass drug administration (MDA) history of the health district? Please provide the number of treatment rounds delivered.

- The use of bivariate and multivariable logistic regression is appropriate. Adjusted odds ratios (AORs) with 95% confidence intervals are reported. Although the Hosmer-Lemeshow test is mentioned, results are not presented and should be included to confirm model fit.

- Ethical clearance was obtained from Addis Ababa University Akililu Lemma Institute of Health Research Institutional Research Ethics Committee, and administrative permission was obtained from the sub-city health and education departments for each selected school. Informed consent procedures are clearly described, and responses were kept confidential and anonymous. Given participants included children as young as 5 years, it is unclear if assent was obtained and what the role of school teachers was in this process; this should be clarified.

**Results:**

-Does the analysis presented match the analysis plan?

-Are the results clearly and completely presented?

-Are the figures (Tables, Images) of sufficient quality for clarity?

Reviewer #1: The analysis presented does not fully align with the expected analysis plan, and some key results appear to be missing or insufficiently detailed. While the tables and figures included are generally relevant, they do not comprehensively capture all aspects of the study findings, particularly those related to infection intensity. The results section would benefit from clearer organization and complete reporting of all analyses outlined in the objectives.

Reviewer #2: The study presents interesting results and corroborates the need for special attention to the health of schoolchildren for the effective elimination of STH. Furthermore, it also presents data on schistosomiasis, which is also a neglected disease waterborne disease whose elimination is between is among the WHO's Sustainable Development Goals. I suggest that the authors present the schistosomiasis data as additional data to the geohelminthiasis data, and not as a total sum.

Reviewer #3: - The results follow the stated analysis plan by estimating prevalence, intensity classification, and identifying associated factors. However, intensity data including low, moderate, and high-intensity infection results for each of the STH species, including S. mansoni, is missing. Supplementary tables referenced, such as intensity data by site, are not included in the main document and should be appended.

- The study population is described as schoolchildren aged 5-18 and their caregivers, but worm prevalence data is not disaggregated by these two groups, as no denominator is provided. The results report the prevalence of S. mansoni, yet the study objectives do not include this. Please disaggregate the prevalence data by subgroup as appropriate.

- Tables should be reformatted for improved clarity, with consistent use of column headers, proper alignment of percentages, and clearer separation of variable categories. It is unnecessary to include district, city, province, and country in the table labels. For example, Table 1 could be reworded from "Socio-demographic and socio-economic characteristics of schoolchildren in Tula Sub-city of Hawassa city, Sidama, Ethiopia in 2025" to simply "Socio-demographic and socio-economic characteristics of study participants." Figure 1 should specifically represent S. mansoni prevalence and not be labeled broadly as "schistosomiasis." Additionally, the label should capture S. mansoni prevalence rather than being generically titled "STH Infection."

- Table 2 is captioned "Multivariate logistic regression associated with soil-transmitted helminths and schistosomes among schoolchildren in Hawella Tula Rural district, Hawassa, Sidama Regional State, Ethiopia, 2025," but no data regarding schistosomiasis is presented. This table is also distorted and difficult to interpret and should be reformatted for clarity.

- Line 207 references Supplementary Table 3, which is not provided in the document and should be included if referenced.

**Conclusions:**

-Are the conclusions supported by the data presented?

-Are the limitations of analysis clearly described?

-Do the authors discuss how these data can be helpful to advance our understanding of the topic under study?

-Is public health relevance addressed?

Reviewer #1: The conclusions are generally consistent with the data presented, but they would be stronger if more explicitly tied to the key findings, particularly regarding infection intensity and associated risk factors. The discussion touches on public health implications, but it could better emphasize how the findings advance understanding of STH transmission dynamics and control strategies in endemic areas. The public health relevance is acknowledged but would benefit from clearer linkage to policy or intervention

Reviewer #2: The conclusions are consistent and the study is relevant to public health.

Reviewer #3: - Conclusions are generally supported by the data, especially the associations between STH and hygiene behaviors, water exposure, and deworming. The need for enhanced intervention was cited but no examples provided.

- Limitations could be expanded to include the low diagnostic sensitivity of Kato-Katz.

- The paper doesn't inform if the WHO threshold for elimination as a public health problem (less than 2% of school-aged children have moderate or heavy intensity infections of any STH species) was achieved for STH. The paper fails to recommend if MDA is required and at what frequency, following WHO guidelines.

**Editorial and Data Presentation Modifications?**

Reviewer #1: Major Revision

Reviewer #2: (No Response)

Reviewer #3: - Correct spelling errors such as “shistosomes” instead of “schistosomes”.

- Ensure all denominators are included in prevalence figures.

- Improve table formatting for clarity and consistency.

- Include missing supplementary table.

- Standardize reference formatting and ensure completeness. For example, Word Health Organization (WHO) is written as "Organization World Health" in references 6,7,9 &10. UNICEF is abbreviated instead of written in full.

**Summary and General Comments**

Reviewer #1: The manuscript presents a timely and relevant investigation into soil-transmitted helminths and schistosomiasis among schoolchildren. The study design, data presentation, and discussion are commendable. However, the entire work would benefit from substantial English language and structural improvement. Additionally, some sections contain redundancy and inconsistent formatting (spacing and punctuation). The authors should also clearly distinguish between STHs and schistosomiasis, as they are biologically and epidemiologically distinct infections. The manuscript would make a valuable contribution to parasitology research.

Reviewer #2: The study presents interesting results and corroborates the need for special attention to the health of schoolchildren for the effective elimination of STH. Furthermore, it also presents data on schistosomiasis, which is also a neglected disease waterborne disease whose elimination is between is among the WHO's Sustainable Development Goals. I suggest that the authors present the schistosomiasis data as additional data to the geohelminthiasis data, and not as a total sum.

Below are some key comments. I am attaching the manuscript with additional comments.

Title: The title suggests reflection on soil-transmitted diseases, which is also observed throughout the text. However, the abstract conclusions already indicate that the study also presents schistosomiasis mansoni, a waterborne disease. Perhaps the title could draw attention to this helminthiasis as well, something like "soil-transmitted helminthiasis and schistosomiasis mansoni" or "transmitted by soil and water," or "helminthiasis in schoolchildren, considering soil-transmitted helminthiasis and schistosomiasis."

Keywords: It is not advisable to repeat keywords that already appear in the title. Prioritize words that will increase the chances of your text being found during reader searches. Some examples that can be used are: Neglected diseases, intestinal parasitosis, schoolchildren's health, STH, Geohelminthiasis, shistosomiasis

Background:

In the background, it is important to clarify the etiological agent that causes each parasitic disease and the mode of infection. Sometimes the authors mention the names of the parasites, in other they refer to the disease by its name, but without having previously stated what causes it (hookworm).

Line 48: “576-740 million with hookworm and 240 million people infected with schistosomiasis (8).” This might be the point where the authors should include schistosomiasis as something in addition to STH:

- Furthermore, schistosomiasis, a waterborne parasitic disease, also presents data that should be considered, with approximately 240 million people infected worldwide.

Lines 217 – 224: The authors present high prevalence values when compared to data from North-West Ethiopia, Birbir town, Southern Ethiopia, and Rwanda. However, it is important to clarify whether these studies include schistosomiasis in the STH data or if this inclusion in the present study may be inflating these numbers.

Reviewer #3: The manuscript presents a well-executed cross-sectional study assessing the prevalence and intensity of soil-transmitted helminthiasis and associated risk factors among schoolchildren in Hawella Tula Rural district, Ethiopia. The topic is of high public health relevance, particularly in sub-Saharan Africa where STH remains endemic. The authors employ standard diagnostic techniques (Kato-Katz), appropriate statistical analyses (bivariate and multivariable logistic regression), and clearly identify modifiable behavioral and environmental risk factors such as poor hygiene, lack of deworming, and exposure to contaminated water. The sample size is adequate and ethical procedures documented. These strengths make the study a valuable contribution to the literature on NTDs and support its potential to inform targeted interventions.

However, the manuscript requires revisions before acceptance. Key issues include the absence of a clearly stated hypothesis, missing denominators in prevalence figures, e.g for S. mansoni prevalence estimation, inconsistent references, and formatting gabs in tables and figures. The discussion section would benefit from clearer structure and deeper integration of findings with global STH control strategies, including WHO’s threshold for elimination (<2% moderate-to-heavy intensity infections). Additionally, the conclusion should offer more specific recommendations for public health action such as frequency of MDA required. Provided these revisions are addressed, the manuscript would meet the standards for publication in PLOS Neglected Tropical Diseases. Therefore, I recommend major revision prior to acceptance.

PLOS authors have the option to publish the peer review history of their article (what does this mean? ). If published, this will include your full peer review and any attached files.

**Do you want your identity to be public for this peer review?** For information about this choice, including consent withdrawal, please see our Privacy Policy .

Reviewer #1: **Yes:** Christianah Oki

Reviewer #2: No

Reviewer #3: No

**Figure resubmission:**
---

## [Decision Letter · Decision Letter 1]

11 Feb 2026

Response to Reviewers
Revised Manuscript with Track Changes
Manuscript

Shaden Kamhawi

co-Editor-in-Chief

Paul Brindley

co-Editor-in-Chief

**Reviewers' comments:**

**Key Review Criteria Required for Acceptance?**

**Methods**

-Are the objectives of the study clearly articulated with a clear testable hypothesis stated?

-Is the study design appropriate to address the stated objectives?

-Is the population clearly described and appropriate for the hypothesis being tested?

-Is the sample size sufficient to ensure adequate power to address the hypothesis being tested?

-Were correct statistical analysis used to support conclusions?

-Are there concerns about ethical or regulatory requirements being met?

Reviewer #1: The objectives to determine prevalence and associated risk factors of STHs and Schistosoma mansoni among schoolchildren are clearly stated. A cross-sectional study is appropriate for estimating prevalence and assessing associations between risk factors. The study focuses on school-aged children in Hawella Tula Rural District, which is the population at risk for STHs and intestinal schistosomiasis. Ethical approval was obtained for the study.

Reviewer #3: There is still no testable hypothesis in the revised manuscript. Below is the authors’ response to my previous query:

"Response: We thank the reviewer for this insightful comment. We agree that a clearly stated hypothesis strengthens the study's framework. In response, we have now added a few sentences at the end of the introduction section to address the testable hypothesis (page 5, lines 93-106) in the revised manuscript, which reads: “Despite various efforts to reduce the burden of soil transmitted helminth (STH) infections and schistosomiasis in Ethiopia, the prevalence of these infections has never fallen below the World Health Organization (WHO) threshold for low prevalence. The goal of this threshold is to reduce morbidity from STH infections in preschool-aged children (pre-SAC) and school-aged children (SAC) by lowering the prevalence of moderate- or heavy-intensity infections to less than 1% in SAC. This ongoing burden is driven by factors including rapid reinfection posttreatment, entrenched environmental exposure risks, and complex socio-demographic determinants. As a result, continuous

monitoring of the socio-demographic, environmental, and lifestyle characteristics associated with an increased risk of STH infections is essential. Tailoring interventions and control strategies to meet the specific needs of each community is critical. This study provides evidence on the prevalence and distribution of STH infections and intestinal schistosomiasis among schoolchildren in southern Ethiopia. It highlights a significant public health challenge and aims to inform targeted interventions for clinicians, public health officials, and policymakers to reduce the burden of STH infections and improve child health.”

I suggested rewording to:

" Despite various efforts to reduce the burden of soil‑transmitted helminth (STH) infections and schistosomiasis in Ethiopia, the prevalence of these infections has never fallen below the World Health Organization (WHO) threshold for low prevalence. This threshold aims to reduce morbidity among preschool‑aged and school‑aged children by lowering the prevalence of moderate‑ or heavy‑intensity infections to less than 1% in school‑aged children. The persistent burden is driven by factors such as rapid reinfection after treatment, entrenched environmental exposure risks, and complex socio‑demographic determinants. Continuous monitoring of socio‑demographic, environmental, and lifestyle characteristics associated with increased STH risk is therefore essential, and tailoring interventions to the specific needs of each community remains critical.

This study provides evidence on the prevalence and distribution of STH infections and intestinal schistosomiasis among schoolchildren in southern Ethiopia. It highlights a significant public health challenge and aims to inform targeted interventions for clinicians, public health officials, and policymakers to reduce the burden of STH infections and improve child health. We hypothesize that specific socio‑demographic and environmental factors, such as household sanitation, water access, and parental education, are significantly associated with the risk of STH infection among school‑aged children".

iv)What criteria were used for school (site) selection?

Response: “convenient sampling techniques used for sub city selection and all three primary schools within the area were included in the study”.

Query: site selection for SCH should be predicated on risk exposure rather than just convenience sampling. Purposeful sampling is accepted on the grounds that factors driving transmission are well defined and characterized in the area where the survey is being conducted.

- Highlight MDA history in the background.

**Results**

-Does the analysis presented match the analysis plan?

-Are the results clearly and completely presented?

-Are the figures (Tables, Images) of sufficient quality for clarity?

Reviewer #1: The study reports prevalence, 95% confidence intervals, and associations between infection and risk factors. The prevalence data and risk factors are reported, but some sections contain repetitive phrasing.

Reviewer #3: No further comments

**Conclusions**

-Are the conclusions supported by the data presented?

-Are the limitations of analysis clearly described?

-Do the authors discuss how these data can be helpful to advance our understanding of the topic under study?

-Is public health relevance addressed?

Reviewer #1: The conclusions about the prevalence of STHs and Schistosoma mansoni, and the associated risk factors (unwashed vegetables, contaminated water, lack of deworming) are directly supported by the study’s results. The recommendations for school-based deworming, hygiene education, and WASH interventions are also reported. The limitations were clearly described. The discussion emphasizes how risk factors contribute to infection and reinfection dynamics. Yes, public health relevance addressed

Reviewer #3: No further comments

**Editorial and Data Presentation Modifications?**

Reviewer #1: Minor Revision

Reviewer #3: Please re-edit all tables (1-4). They still appear quite distorted. Ensure the figures are numbered in sequence.

**Summary and General Comments**

Reviewer #1: The manuscript addresses an important public health issue by assessing the prevalence and risk factors of soil-transmitted helminths (STHs) and Schistosoma mansoni among schoolchildren in southern Ethiopia. The study design, data presentation, and discussion are commendable. However, the entire work would benefit from substantial improvements in the English language and structure. Additionally, some sections contain redundancy and inconsistent formatting (spacing and punctuation). Several sentences are run-on or contain subject-verb agreement errors. Some paragraphs, especially in the Results and Discussions, contain repetitive phrasing and abrupt transitions between STHs and schistosomiasis. References and statistical reporting (percentages with 95% CI) should be uniform. The manuscript would make a valuable contribution to parasitology research.

Reviewer #3: (No Response)

PLOS authors have the option to publish the peer review history of their article (what does this mean? ). If published, this will include your full peer review and any attached files.

**Do you want your identity to be public for this peer review?** For information about this choice, including consent withdrawal, please see our Privacy Policy .

Reviewer #1: No

Reviewer #3: No

**Figure resubmission:**

**Reproducibility:** To enhance the reproducibility of your results, we recommend that authors of applicable studies deposit laboratory protocols in protocols.io, where a protocol can be assigned its own identifier (DOI) such that it can be cited independently in the future. Additionally, PLOS ONE offers an option to publish peer-reviewed clinical study protocols. Read more information on sharing protocols at https://plos.org/protocols?utm_medium=editorial-email&utm_source=authorletters&utm_campaign=protocols

---

## [Editor Report · Decision Letter 2]

3 Mar 2026

Dear Mr Bonja,

We are pleased to inform you that your manuscript 'Prevalence, Infection Intensity, and Risk Factors of Soil-Transmitted Helminthiasis and Intestinal Schistosomiasis among Schoolchildren in Southern Ethiopia' has been provisionally accepted for publication in PLOS Neglected Tropical Diseases.

Best regards,

Maria Y Pakharukova, Ph.D., D.Sc.

Academic Editor

Eva Clark

Section Editor

Shaden Kamhawi

co-Editor-in-Chief

Paul Brindley

co-Editor-in-Chief

---

## [Editor Report · Acceptance letter]

Dear Mr Bonja,

We are delighted to inform you that your manuscript, "Prevalence, Infection Intensity, and Risk Factors of Soil-Transmitted Helminthiasis and Intestinal Schistosomiasis among Schoolchildren in Southern Ethiopia," has been formally accepted for publication in PLOS Neglected Tropical Diseases.

Best regards,

Shaden Kamhawi

co-Editor-in-Chief

Paul Brindley

co-Editor-in-Chief
